# In-shoe plantar shear stress sensor design, calibration and evaluation for the diabetic foot

Athia H. Haron[1], Lutong Li[1], Jiawei Shuang[1], Chaofan Lin[1], Helen Dawes[2], Maedeh Mansoubi[2], Damian Crosby[1], Garry Massey[2], Neil Reeves[3], Frank Bowling[4], Glen Cooper[1], Andrew Weightman[1]*

1 Department of Mechanical, Aerospace and Civil Engineering (MACE), University of Manchester, Manchester, United Kingdom, 2 Medical School, NIHR Exeter BRC, University of Exeter, Exeter, United Kingdom, 3 Musculoskeletal Biomechanics and Research in Science and Engineering faculty of Manchester Metropolitan University, Manchester, United Kingdom, 4 Manchester University NHS Foundation Trust within the Departments of Diabetes and Vascular Surgery, Manchester, United Kingdom

* Andrew.Weightman@manchester.ac.uk

**Data Availability Statement:** All relevant data are available from the Mendeley Data database (DOI 10.17632/pcggh2rzm3.1). Only raw anonymised data can be shared due to GDPR restrictions from HRC ethics committee.

## Abstract

Plantar shear stress may have an important role in the formation of a Diabetic Foot Ulcer, but its measurement is regarded as challenging and has limited research. This paper highlights the importance of anatomical specific shear sensor calibration and presents a feasibility study of a novel shear sensing system which has measured in-shoe shear stress from gait activity on both healthy and diabetic subjects. The sensing insole was based on a strain gauge array embedded in a silicone insole backed with a commercial normal pressure sensor. Sensor calibration factors were investigated using a custom mechanical test rig with indenter to exert both normal and shear forces. Indenter size and location were varied to investigate the importance of both loading area and position on measurement accuracy. The sensing insole, coupled with the calibration procedure, was tested one participant with diabetes and one healthy participant during two sessions of 15 minutes of treadmill walking. Calibration with different indenter areas (from 78.5 mm$^2$ to 707 mm$^2$) and different positions (up to 40 mm from sensor centre) showed variation in measurements of up to 80% and 90% respectively. Shear sensing results demonstrated high repeatability (>97%) and good accuracy (mean absolute error < ±18 kPa) in bench top mechanical tests and less than 21% variability within walking of 15-minutes duration. The results indicate the importance of mechanical coupling between embedded shear sensors and insole materials. It also highlights the importance of using an appropriate calibration method to ensure accurate shear stress measurement. The novel shear stress measurement system presented in this paper, demonstrates a viable method to measure accurate and repeatable in-shoe shear stress using the calibration procedure described. The validation and calibration methods outlined in this paper could be utilised as a standardised approach for the research community to develop and validate similar measurement technologies.

**Funding:** This work was partially funded by Engineering and Physical Sciences Research Council (EPSRC) grant number EP/W00366X/1.

**Competing interests:** The authors have declared that no competing interests exist.

## Introduction

Diabetic foot ulceration (DFU) affects 15–25% of people with diabetes at some point in their lifetime [1] and has a high social and economic cost with countries like the UK spending approximately £1 billion annually [2]. Worldwide the prevalence of diabetes is rising, and it is predicted that 552 million people will have the condition by 2030 [3]. Measurement of plantar normal stress and plantar shear stress has shown the potential to predict DFU risk [4, 5]. However, whilst commercial systems are available to measure normal plantar stress in-shoe there are no commercially available in-shoe plantar shear stress measurement systems. Shear stress has been directly measured during barefoot gait using mechanical sensor arrays coupled with resistive or capacitive sensors [6–8], utilising piezoelectric materials and their charge outputs [9] and through a variety of optical methods including polycarbonate arrays [6], optical bend loss [7] and laser interferometry of bi-refringent films [8]. Perry et al. [10] used an array-based device [11] to study bunching and stretching of adjacent plantar tissue and they found that tissue stretching from shear stress was the predominant mechanism. They report that peak shear stress and peak plantar pressure occur in the same place in 50% of cases, but actually occur at different times, which is contradictory to results reported by other researchers [12]. Contradictory results are typical from these studies using custom-built shear stress measurement devices due to the relatively low numbers of participants with diabetes tested in the trials, with typical sample sizes of ten. All these measurement methods are bespoke devices and only a handful of foot-to-floor shear stress measurement devices exist worldwide. Larger scale studies with matched control groups are required to provide firm conclusions on plantar surface shear stresses experienced by people with diabetes.

Shear stress measurement is further complicated as all diabetic patients are strongly advised to walk using footwear (and never barefoot), therefore, to understand the shear stresses induced on the plantar surface, in-shoe shear stress measurement must be taken. Although direct shear stress measurement is important in DFU risk management, future use of artificial intelligence methods [13, 14] may enable risk management with current measurement technologies.

In-shoe plantar shear stress is difficult to measure and reported measurements vary widely, for example, measurements of shear stress on the 1st metatarsal head varied from 16 kPa [15] to 140 kPa [5] in healthy participants. Therefore, either there is widespread inter-participant variability and/or there are mechanisms which cause errors for in-shoe shear stress measurement. Measurement error has been widely reported for in-shoe normal stress systems with causation linked to sensor wear and calibration [16, 17]. Specifically, calibrating with similar load ranges to those desired to be measured improved accuracy by up to 20 times [16] and accuracy was reduced when smaller areas of loading were applied [17]. It is likely that similar calibration issues will affect in-shoe shear stress sensor measurement accuracy. Researchers have made excellent progress in developing novel in-shoe plantar shear stress measurement systems; however, they have not yet fully considered the implications of calibration methods on measurement accuracy [4, 5]. The choice of indenter area of loading, shape and location is also an important consideration for accurate and reliable sensor calibration; despite this, to the authors' knowledge this has not been investigated and reported in the literature. A key principle in calibration is that the applied loading should be a good representation of the real-world scenario. In the context of plantar foot mechanics, and for example the metatarsal heads, there is variation in the magnitude of loading, area of loading, shape and potentially the location of the bone in relation to the sensor. This paper presents the design and evaluation of an in-shoe shear stress sensor and considers the impact of calibration on measurement accuracy.

## Methods

This paper describes a sensor system design and conducts a performance investigation. Three investigations were conducted: calibration investigation, loading profile comparison and sensor validation. These investigations and how they relate to one another are shown in Fig 1.

### Sensor system design

**Sensing principle.** Coulomb's law of friction describes frictional force being proportional to reaction load. In the case of shear sensing insoles this means that there can be no shear stress (friction) without normal stress (reaction load) and that the magnitude of associated shear stress will always be less than that of normal stress. Like most other shear sensors in the

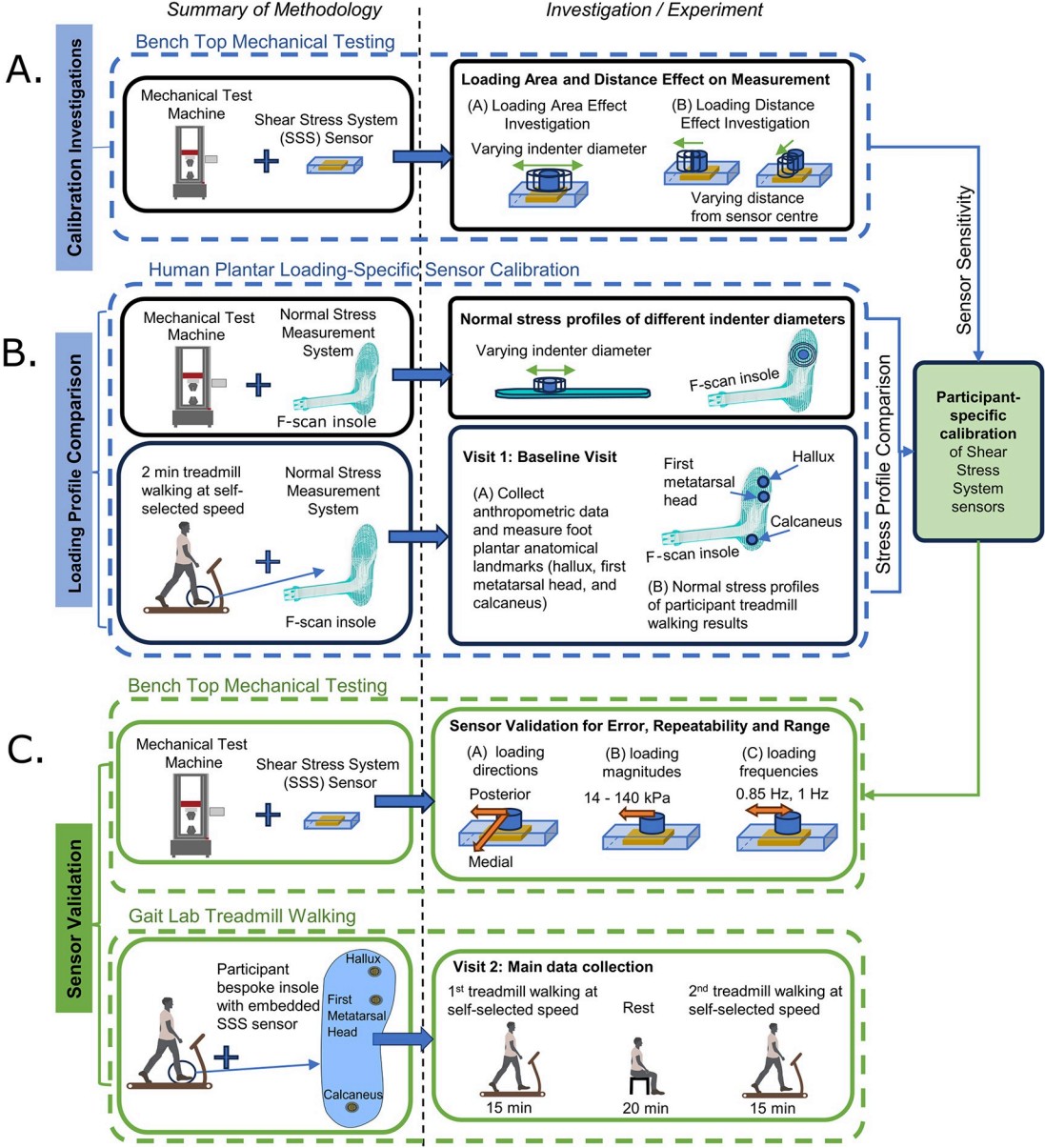

**Fig 1. Summary of methodology for investigations into sensor calibration, loading profile and sensor validation.**

literature [5, 15, 18–22] the shear sensor is embedded in a hyperelastic or viscoelastic, isotropic incompressible elastomer, as opposed to a discrete sensor placed on the insole or isolated from the main body of the insole material.

Fig 2A shows a cylindrical section of elastomer insole with cross-sectional area, A, containing a strain gauge orientated in the shear plane and a normal stress sensor with sensor readings in mV, S, and N, respectively. The material properties (stress-strain relationship) for the silicone are non-linear but can be approximated as three linear regions (low: $\leq \varepsilon_1 = 0.04$ *strain*; medium: $\leq \varepsilon_2 = 0.115$ *strain*; high: $> \varepsilon_2 strain$); see Fig 2B. The strains for the three linear regions were determined from the stress-strain curve of the silicone under compressive loading at the target stresses of 14 kPa (low), 70 kPa (medium) and 140 kPa (high). Stress-strain relationships for normal compressive loading are given by Eq 1, where $C_{medium}$, and $C_{high}$ are negative intercepts in units of pascal ($C_{low} = 0$) and $E$ is the gradient in Pascal.

$$\sigma_N \begin{cases} E_{low} \cdot \varepsilon_N - C_{low} = E_{low} \cdot \dfrac{\Delta t}{t_0} ; for 0 \leq \varepsilon < \varepsilon_1 (low) \\[2ex] E_{medium} \cdot \varepsilon_N - C_{medium} = E_{medium} \cdot \dfrac{\Delta t}{t_0} - C_{medium} ; for \varepsilon_1 \leq \varepsilon > \varepsilon_2 (medium) \\[2ex] E_{high} \cdot \varepsilon_N - C_{high} = E_{high} \cdot \dfrac{\Delta t}{t_0} - C_{high} ; for \varepsilon > \varepsilon_2 (high) \end{cases} \quad (1)$$

Fig 2C shows the section being loaded with a normal force which creates a reduction in thickness but an increase in diameter described by Eq 2 (assuming constant volume) which

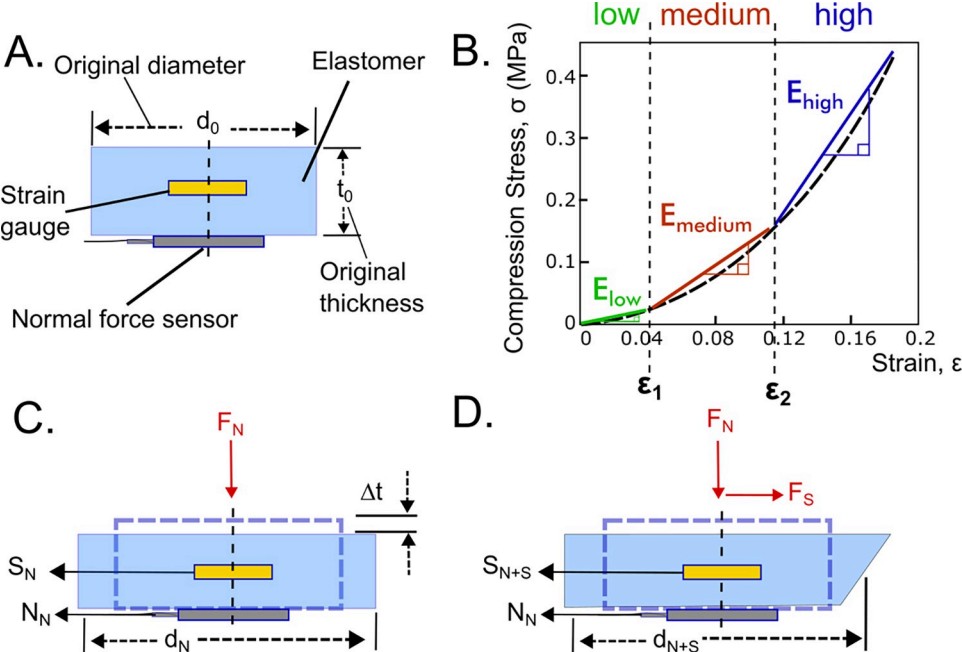

**Fig 2. Sensing principle.** [A] Cylindrical section of elastomer containing strain gauge and normal force sensor [B] Stress-strain curve of the elastomer under compression stress. Linear approximations for deformation were made for three regions of the curve (low, medium, and high stress magnitudes), sectioned by the compressive strains $\varepsilon_1$ and $\varepsilon_2$, with corresponding gradient E used for calibration. [C] Cylindrical section deformed by normal force only. [D] Cylindrical section deformed by both normal and shear forces.

gives sensor readings $S_N$ and $N_N$, which are signal voltage measurements (mV) for the shear stress and normal stress respectively, described by Eqs 3 and 4 where $k_N$ and $k_S$ are constants (sensor gains) determined by experiment with units Pa/mV and strain/mV respectively (other equation parameters defined in Fig 2 with SI units).

$$d_N = \sqrt{\frac{d_0{}^2 t_0}{t_0 - \Delta t}} \tag{2}$$

$$N_N = \frac{F_N}{k_N \cdot A} = \frac{4F_N}{k_N \pi d_0{}^2} \tag{3}$$

$$S_N = \frac{d_N - d_0}{k_S \cdot d_0} \tag{4}$$

Fig 2D shows applied loading from both normal and shear force giving a sensor reading $S_{N+S}$ and $N_N$ for the shear stress and normal stress respectively. The applied shear stress, $\sigma_S$, can be determined from Eq 5 and Eq 1 (assuming an isotropic material) which requires measurements from the normal stress sensor, $N_N$, to decouple the effect on the strain gauge from normal force (where $i = low$, $medium$ or $high$).

$$\sigma_S = \frac{F_S}{A} = E_i \cdot k_S \cdot \left( S_{N+S} - S_N \right) - C_i \tag{5}$$

**Shear stress sensor design.** The shear stress sensing system primarily consists of the strain gauge rosette, a normal stress sensor, and the flexion stiffener and load concentrator; here on in referred to as the 'shear stress system sensor' or 'SSS sensor'. A 3-element strain measuring rosette (1-RY81-3/120, Hottinger Bruel & Kjaer UK Ltd, Royston, England) was chosen for the shear stress sensor (Fig 3A) arranged in rectangular 0°-45°-90° directions to allow for calculation of resultant shear in both the anterior-posterior (AP) and medial-lateral (ML) directions. The sensor was then embedded in silicone (Sil A50 Smooth- Sil Addition Cure silicone, Smooth-On Inc. Macungie, USA). To assemble the sensor, the first 2mm silicone base layer was poured into a custom 3D printed square mould with dimensions of 20 x 20 x 4 mm (width x length x height). After curing the surface was cleaned and the strain gauges were soldered to 2-core 2.8 mm² external diameter shielded wires (JY-1060, Pro-Power by Newark, Chicago, USA). The strain gauges were then placed on the surface of the silicone using a custom 3D printed jig with tabs and bolts to align the strain gauges in the correct angular position. A thin second layer of silicone (approximately 0.5 mm thick) was then poured and allowed to fully cure, the jig was then removed and a final layer of silicone was poured on top to give a total thickness of 4 mm. A 15 mm diameter, 0.8 mm thick phenolic sheet material flexion stiffener and load concentrator was placed at the center of the sensor assembly and the top layer of silicone was then allowed to cure. The full assembly of the sensor is shown in Fig 3B and 3C.

As mentioned in the sensing principle, the shear stress is obtained from Eq 5, however, for the SSS sensor to measure both AP and ML shear stress, orientation of the strain gauges needs to be considered. From the configuration shown in Fig 3A for stress measurements calculated from strain gauges A, B and C the shear stress is given by Eqs 6 and 7.

$$\sigma_{AP\_Shear} = \sigma_C \sin \theta_{BC} - \sigma_A \sin \theta_{AB} \tag{6}$$

$$\sigma_{ML\_Shear} = \sigma_B + \sigma_C \cos \theta_{BC} + \sigma_A \cos \theta_{AB} \tag{7}$$

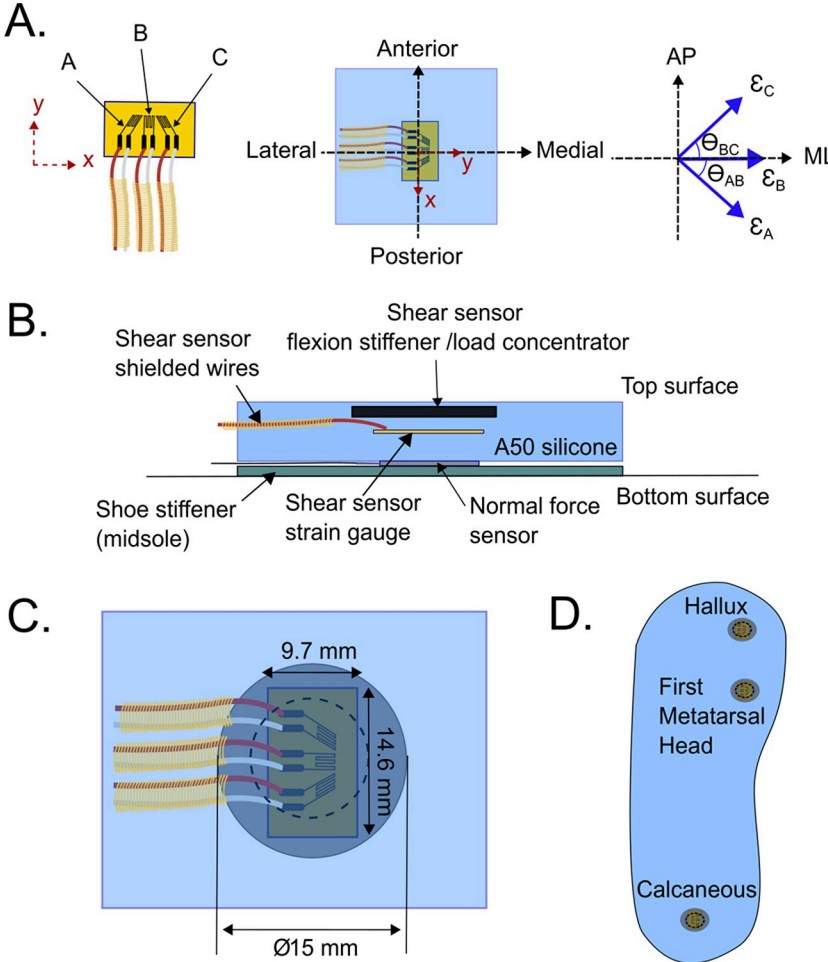

**Fig 3. Shear stress system (SSS) sensor and placement in the sensing insole.** [A] Configuration of the strain rosette in the sensor with three strain gauges arranged at 0˚– 45˚ - 90˚, and the relationship between the local strain axes and the global applied shear direction axes (Medial-Lateral, ML, and Anterior-Posterior, AP). [B] Section view of the SSS sensor. [C] Top view of the SSS sensor and its dimensions. [D] Locations of SSS sensors in the sensing insole.

Where $\theta_{AB}$ and $\theta_{BC}$ are the angles between the individual strain gauges in the rosette, which were at 45˚.

**Shear stress sensor number, placement, and integration.** Key DFU risk areas, accounting for at more than one-third of DFU cases are the calcaneus, first metatarsal head and hallux areas of the foot [23–26], so placement of the SSS sensors in the insole was in these three locations. To maximize accuracy of the measured sensing data, all sensors were anatomically matched to the participant. This was achieved through a 'palpation and marking paper' approach in which a healthcare professional identified the bony landmarks of the foot, marked these areas on the foot surface with skin-safe marker, and the participant stands on the paper to transfer the markings. These markings were then used to ensure SSS sensors were correctly located on the silicone insole, with the sensor x-axis aligned with the anterior posterior direction. The signal wires were laid out from the SSS sensor in the ML direction to reduce fatigue loading from flexion during gait. A 1–2 mm depth of silicone was then poured and cured before a further layer of silicone was poured and cured to make a total insole thickness of 5 mm to complete the insole, as shown in Fig 3D. Three normal stress sensors (A301 FlexiForce 0-44N, Tekscan Inc., Norwood, Massachusetts, USA) were then secured to the bottom of the

insole with silicone glue (Permatex 80050 Clear RTV Silicone Adhesive Sealant, Permatex, Illinois Tool Works Inc., Solon, Ohio, USA) with their center coincident with the SSS sensors.

**Data acquisition system (DAQ) and signal processing.**    A Teensy 4.1 32-bit microcontroller (PJRC, Portland, Oregon, USA), ARM Cortex-M7 processor, with clock speed of 600 MHz and integrated SD storage card, was used to collect and store the voltage readings from the SSS sensors (Fig 4B). Flexiforce normal stress sensors were connected via a 10 kOhm circuit divider to analog inputs, whilst shear sensing strain gauges were amplified using a 24-bit high-precision analog-to-digital amplifier (HX711 ADC, HALJIA, Zhongai, China) then routed to digital inputs of the microcontroller. All signals were collected at a sampling rate of 80 Hz. Data was logged to the 16GB SD card and streamed via an ESP8266 UART WiFi adapter (Espressif Systems, Shanghai, China) to allow for continuous monitoring. Power was supplied to all components via 3V and 5V power rails from the microcontroller, sourced from an external 3.7V 3500mAh Lithium Polymer battery (LP104567, EEMB, Moscow, Russian Federation) that was regulated through a linear regulator (LDO, B08HQQ32M2, DollaTek, Hong Kong, China). For both left and right foot measurements, two identical systems were used to collect the measurements, and placed on a custom, adjustable neoprene fitness belt (Frienda, China), during walking trials (Fig 4).

A custom MATLAB (The Mathworks Inc., Natick Massachusetts, USA) script was used to parse and analyse the data collected. The data was minimally pre-processed before finalized into calibrated stress measurements. This pre-processing stage included removing only obvious outliers (which accounted for up to 0.05% of the measurement data if present). This was made using the *filloutlier* function with the 'quartile' outlier detection option: 'quartiles' outliers which were elements more than 1.5 interquartile ranges above the upper quartile (75 percent) or below the lower quartile (25 percent)) and correcting DC offsets. Data from each foot were analyzed separately.

## Calibration investigation: Bench top mechanical testing

**Experimental setup and test method.**    To investigate the effect of calibration on the sensor's performance, both shear and normal force were applied to the SSS sensor insole (summarised in Fig 1A). A uniaxial mechanical testing machine (Instron 5982K2680 100kN 350˚C, 500N load cell, Instron ® Norwood, Massachusetts, USA) applied and measured shear force using a bespoke shear stress rig through an indenter of area, A, shown in Fig 5A. A normal reaction force was applied through a screw thread to the indenter to facilitate frictional shear stress application. Measurement of normal reaction force was through a load cell and ADC (ADN1903027, 196.2 N Weight Sensor Load Cell, Haljia, China) capturing data at 80Hz using an Arduino (Arduino Mega 2560 Rev3, Arduino, Somerville, MA, USA). For pure normal stress loading calibration, the insole was placed flat on a plate in the uniaxial testing machine fitted with a large compression platen on the bottom and an indenter with a specific area, A, applying compression force from the top, shown in Fig 5B.

**Sensor loading area investigation.**    To evaluate the effect of indenter area, A, five flat ended cylindrical indenters with diameters of, 10, 15, 20, 25, 30 mm were used to load the SSS sensor at its center. While studies have shown that there is a difference between various indenter shape loading profiles and the corresponding mechanical responses of the material [27, 28], we determined that the normal stress distribution that was measured at the surface of the SSS sensor was similar for both flat and rounded indenter profiles. The only notable difference was the size of the normal stress distribution, as a flat indenter covered a larger area than the rounded indenter of the same diameter. Thus, choosing a flat indenter of a smaller size gave the same loading results as a larger rounded indenter.

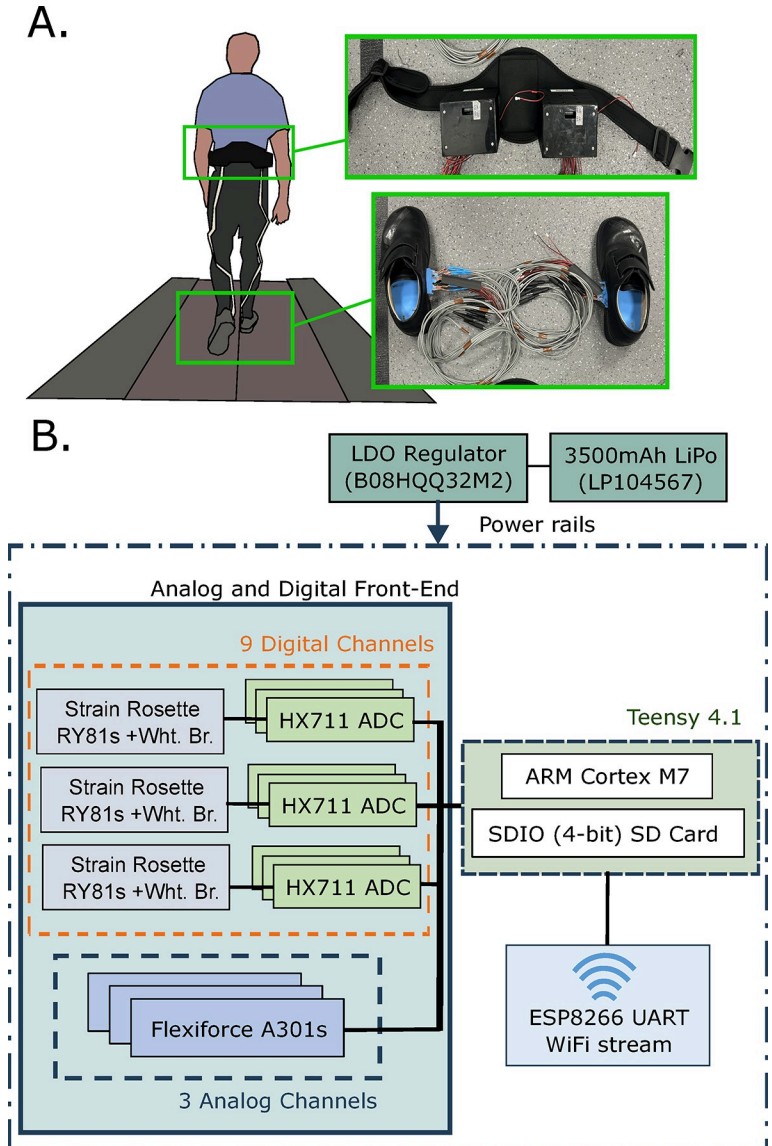

**Fig 4. Treadmill walking setup and data acquisition.** [A] Participant walking on a treadmill with the sensor insole system. The data acquisition system (DAQ) was attached to a belt, and each insole (left and right foot) has a separate but identical DAQ input. [B] Block diagram of the DAQ system, collecting data at 80 Hz.

The tests applied a cyclic shear force with a 1 Hz triangular waveform pattern ranging from 0 to 50 N in combination with a constant normal stress of 140 kPa through all the indenters. SSS Sensor output signal, $S_{N+S}$, in mV was measured for each of the loading areas.

**Sensor loading location investigation.**   Ideally a sensor would be co-located with the anatomical part applying the load, however, this may not always be practically possible so an understanding of the relationship between the location of the SSS sensor, the location of the applied loading and the accuracy of measurement is required. To investigate the effect of loading location, twelve loading locations were chosen, six in the anterior direction and six in the lateral direction both measuring 0, 10, 15, 20, 30, 40 mm from the center of the shear stress sensor. Loads were applied in both the medial or posterior direction respectively. Cyclic loading was applied to the SSS sensor insole of the same characteristic as the area of loading

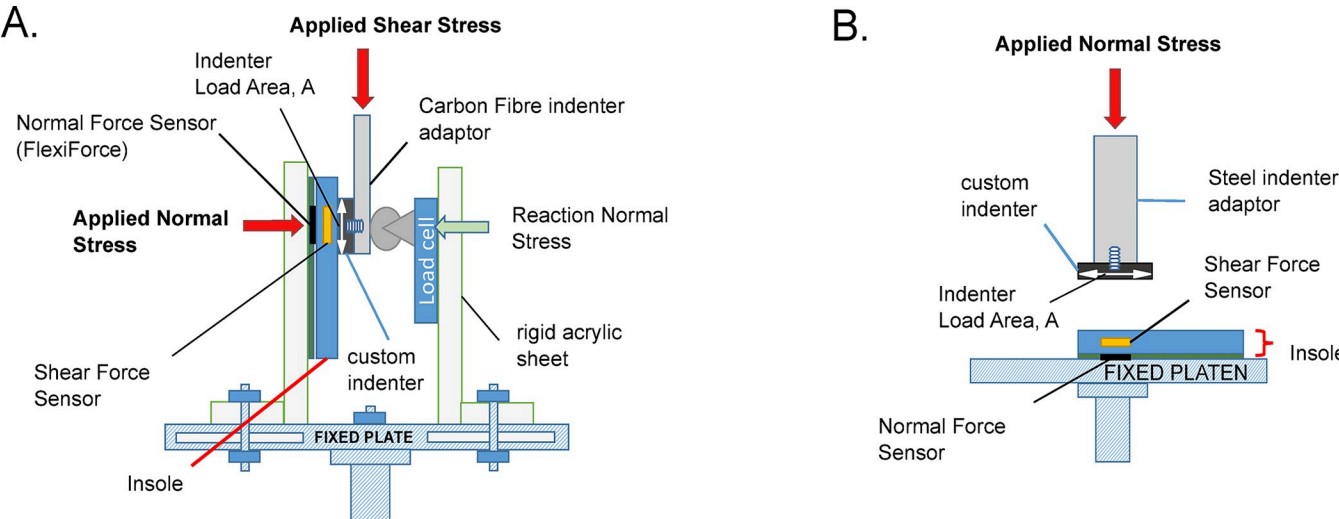

**Fig 5. Calibration setup.** [A] Custom shear stress rig made of rigid 10 mm acrylic sheet plates which applied the force of the mechanical testing machine as a shear force onto the insole. The shear stress was calculated using the applied force and area of the custom indenter. The indenter's compressive stiffness was 30.1 MPa, ~12 times stiffer than the silicone sensor of 2.5 MPa. [B] Custom normal stress calibration setup where the insole was placed on a compression platen.

investigation (see 'Sensing loading area investigation' section). SSS Sensor output signal, $S_{N+S}$, in mV was measured for each of the loading locations.

## Loading profile comparison: Human plantar loading specific sensor calibration

**Comparison of normal stress profiles.** Shear loading application area and location affect strain measurements, so it is important to consider plantar stress loading from the human foot. During walking plantar stress is dependent on many factors including foot size and anatomy, weight, morbidity and walking patterns, all of which are different between participants. From the sensor calibration investigations in the results section, we can see that (i) loading location and (ii) loading area may affect the output of the SSS sensor so these must be considered during calibration.

i. Loading location variation can be removed by placing the SSS sensors at personalised anatomical locations in the insole, which is the approach we have taken.

ii. Loading area variation can be controlled through calibration. This was determined through a comparison and matching of normal stress loading profiles of the specific participant's foot anatomy with bench top mechanical test experiments involving various loading area sizes (flat cylindrical indenters).

To capture the plantar normal stress loading profiles of our participants, in the SSS sensor locations of the calcaneus, first metatarsal head and the hallux, we conducted measurements in-shoe during a two-minute treadmill walk using an F-scan insole (Tekscan Inc., Boston, USA) coupled with a non-instrumented insole of the same material properties and thickness as our designed insole. Then the test rig (Fig 4B) was used with 15, 20, 30 and 40 mm diameter indenter sizes to load the silicone insole from 0 to 250 N (to simulate a normal stress range up to 1400 kPa, which is comparable to the 1000–1900 kPa normal plantar stresses during gait reported in the literature [29, 30].

Measurements of plantar normal stress distribution were captured with the same F-scan and insole used with the participants. To simulate the different foot structures, we adjusted the diameter of cylindrical indenters (15, 20, 30 and 40 mm), which were based on the ranges of average anatomical dimensions of the hallux, metatarsal head, and calcaneus bones [31–36], see results and discussion 'Human plantar loading consideration for sensor calibration' section. An illustrated summary of this investigation can also be found in Fig 1B.

**Statistical analysis as a method for calibration indenter choice.** Comparisons were made between the participant's mean normal stress profiles with the bench top test rig results (gait data averaged over 20 gait cycles from three different sensing locations hallux, first metatarsal head, and calcaneus, bench top test rig results for 15, 20, 30 and 40 mm indenter diameters). Magnitudes of both results were scaled to have a maximum unity magnitude to enable comparison. The normal stress profiles (normal stress vs displacement across anatomical location) were collected along a 2D cross section of 40 mm in length across the foot-width of loaded area (see results and discussion 'Human plantar loading consideration for sensor calibration' section). Calibration indenter diameters for the hallux, first metatarsal head and calcaneus locations were chosen based on either the highest $R^2$ value from a multiple linear regression between the gait measures and the test rig measures or the maximum measurement sensitivity area of the SSS sensor (see results and discussion 'Sensor calibration' section).

## Sensor validation: Bench top mechanical testing

The following section describes the sensor validation, as summarised in Fig 1C. A 30 mm diameter indenter was used to calibrate the SSS sensor, as this was determined to be the maximum sensing area of the sensor (see results and discussion 'Sensor calibration' section). This was achieved through a series of mechanical tests detailed in Table 1, with shear stresses applied in both ML and AP directions and conducted at 1Hz, to simulate average walking speed frequency.

The shear stress magnitudes chosen for low, medium, and high levels were 10%, 50% and 100% of the 140 kPa maximum in-shoe plantar shear stress reported in the literature respectively [37]. This enabled calculation of the calibration parameters coefficients $E_{low}$, $E_{high}$, $C_{medium}$ and $C_{high}$, according to Eq 1.

To validate the calibrated SSS sensor, a shear stress of 70 kPa with a normal stress of 125 kPa was applied in both the ML and AP direction at 0.8 Hz. Additionally, a shear stress was also applied in the 45˚ direction (14 kPa shear stress, 28 kPa normal stress at 1Hz).

Two measurements of error were made. The first was an overall mean absolute error (MAE), which is the mean of the difference between the measurement from the test rig and the calibrated SSS sensor measurement (in kPa). The second was peak error, measured as the percentage error at peak loads between the applied measurement from the test rig and the calibrated SSS sensor measurement. Peak values of measured shear stress were taken from 10 cycles and a standard deviation was calculated. Repeatability was calculated from the SSS sensor measurements as the standard deviation of the peak plantar stresses divided by the mean of

**Table 1. Three conditions for mechanical bench top testing.**

| Testing condition | Low | Medium | High |
|---|---|---|---|
| **Cyclic shear stress*** | 14kPa | 70kPa | 140kPa |
| **Constant normal stress** | 28kPa | 140kPa | 280kPa |

*Triangular at 1 Hz with normal stress of at least twice the maximum of the applied shear stress is applied to prevent slip

the peak plantar stress, presented as percentage (e.g. a mean peak measurement of 100 kPa and a standard deviation of those peak measurements at ± 10 kPa, would result in (10/100) x 100% = 10% deviation from the peak value, and thus 90% repeatability).

## Sensor validation: Gait lab treadmill walking

To further validate the sensors, a gait laboratory treadmill walking test was performed on a single anthropometrically matched healthy participant and a single participant with diabetes (both male and 45 years old, weighing 88 kg and 75 kg, height of 1.75 and 1.66 m, EU shoe size 44 and 42, weight per insole area 32 kPa and 35 kPa, walking speed 0.92 ms$^{-1}$ and 0.95 ms$^{-1}$ for the healthy participant and participant with diabetes respectively). The study received approval from the NHS Health Research Authority and Health and Care Research Wales (HCRW) Ethics Committee (REC reference: 22/NW/0216), and all participants provided written consent. Trial Registration number: NCT05865353. Participants were recruited between 1$^{st}$ November 2022 till 30$^{th}$ May 2023. Data collection was conducted in two parts (1) baseline visit and (2) main data collection, Table 2.

**Baseline visit.**   Anthropometric data was collected, and anatomical landmarks determined using the 'palpation and marking paper' method described in the 'Shear stress sensor number, placement and integration' section. The participants conducted a 2 minute treadmill walk while wearing a pair of silicone insoles, made from the same materials and dimensions as the sensor insole but without active sensors, and a pair of F-Scan pressure sensing insoles, in a prophylactic shoe (Sponarind 97308, Finn Comfort Inc. Hassfurt, Bavaria, Germany), designed with shock-absorbing properties and a larger volume, ideal for people with diabetes. Normal stress data was collected using the F-scan insoles, at a self-selected gait speed to determine normal plantar stress profiles (results of which were used for the comparison of normal stress profiles, in 'Human plantar loading specific sensor calibration' section). Table 2 shows the participant data collected during the baseline visit.

**Main data collection.**   The participants returned for the main data collection where they were asked to wear the sensing insole in the specialist diabetic shoe. They then walked twice on a split belt treadmill with integrated force plates (M-Gait, Motek Medical BV, Amsterdam, Netherlands) for 15 minutes at their self-selected speed (see Table 2).

**Data analysis: Shear stress gait measures and repeatability.**   Mean and standard deviation of peak shear stress and peak normal stress measurements were extracted from 20 gait cycles measured by the sensing insole. Measurement repeatability was determined and comparisons, between the two walking periods within each individual walking session (start, middle, and end). We collected statistical data for both plantar shear stress and normal stress measurements to perform inter-participant comparisons. These included statistics for Plantar Stresses (Normal, AP Shear, and ML Shear) across all three sensor areas, encompassing mean

**Table 2. Participant visits and activity.**

| Visit | Purpose | Activity |
|---|---|---|
| **Baseline Visit** | Collect anthropometric data, foot anatomical landmarks, baseline walking plantar normal stress profiles during gait data with mock-up insoles* and shoe. | 1. Physio palpation of the foot to determine anatomical landmarks for the manufacturing of bespoke insoles. <br> 2. Two-minute walking on treadmill with force plates and F-scan insole system. |
| **Main Data Collection** | Collect in-shoe normal and shear plantar stress during gait | 10 minute sit -> 5 minute stand -> **Walk 1:** 15 minute treadmill walk (self-selected speed) -> 20 minute rest -> 5 minute stand -> **Walk 2:** 15 minute treadmill walk (same self-selected speed) |

*Without active sensors but made with the same materials as sensor insole

values, standard deviations, peak stresses, and variability (or percentage difference) of measurements within the 15-minute treadmill walk (intra-walk) and between two treadmill walks (inter-walk).

## Results and discussion

### Sensor calibration

Shear stress measurement accuracy is affected by the calibration method. Specifically, the shear stress sensor measured output signal decreases exponentially with both increasing loading application area, and increasing loading distance away from sensor center, see Fig 6. The results in Fig 6A show that the measured output decreases by ~80% from 1.5 mV to 0.3 mV,

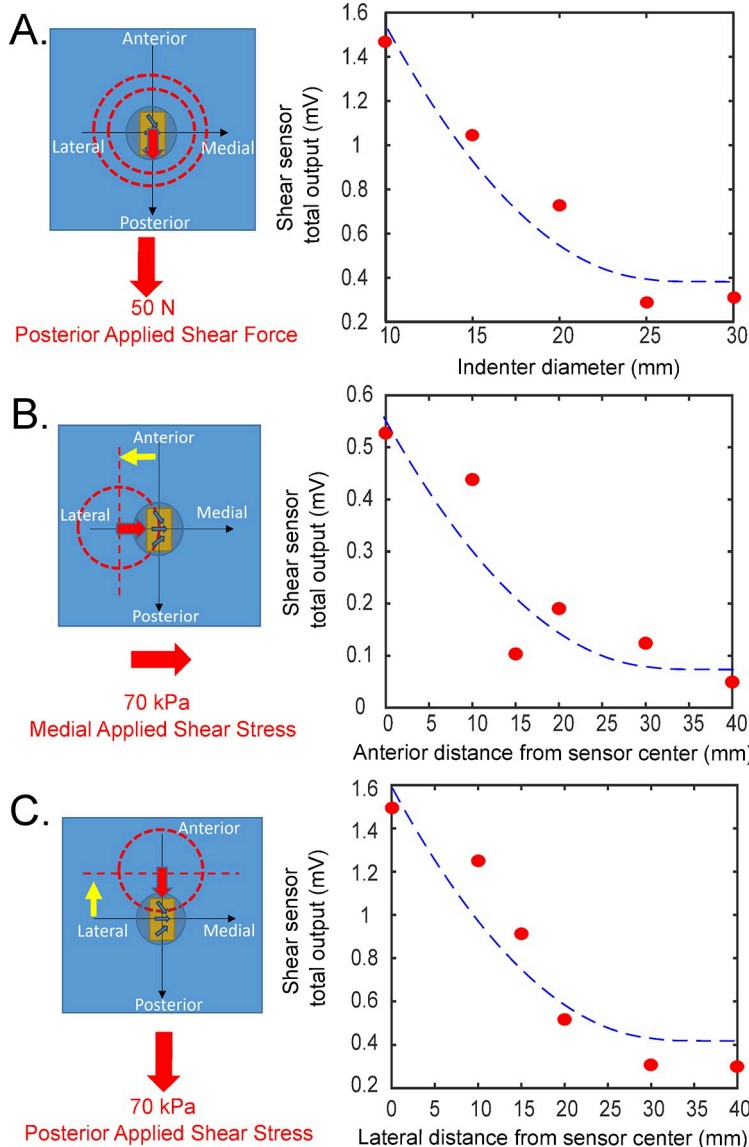

**Fig 6. Calibration investigation results.** Mean peak signal of shear stress (SSS sensor) total output (mV) from 10 cyclic triangular loading. [A]—Effect of area of loading on SSS sensor measured outputs. [B, C]- Effect of location of loading on SSS sensor output for medial and posterior respectively.

for a calibration loading application area of 10 mm diameter to 30 mm diameter respectively. This means that if the sensor was calibrated for the smaller 10 mm area and a larger 30 mm diameter load was applied, the measurements would be underestimated by 80%. Likewise, calibrating for a larger area, and applying load for a small area will greatly overestimate the measurements. Increasing the loading application area increases the area over which the force is distributed over the sensor, thus more of the loading is applied away from the center of the shear stress sensor. From the results shown in Fig 6B and 6C the location of loading application also reduces sensor sensitivity. All this means that the shear stress sensor will only be able to measure accurately if the calibration loading area matches the desired measurement loading application area (or are reasonable similar areas).

Fig 6B and 6C show the influence of loading location on SSS sensor measurements for the same applied loading area (25 mm diameter indenter). As expected, the SSS sensor measurement for both Anterior-Posterior (AP) and Medial-Lateral (ML) shear loading decreased as the loading distance moved away from the sensor center. This is due to a decrease in deformation of the shear stress sensor as the loading is applied further away from the sensor center. However, it is important to note that there was still a measurable signal at these distances as they are not yet relatively far away from the sensor. This means that measured shear stress from an embedded sensor will not just be from the coincident anatomical location but also have a contribution from adjacent and other relatively close anatomies (e.g. first metatarsal head located sensor may be measuring shear stress contribution from the second metatarsal head). This is due to material coupling, which is that stress applied in one area of the material, in this case the silicone insole, will stress surrounding areas of the material. The implication is that the shear stress sensor will provide more accurate measurements if the loading application location is coincident with the centre of the sensor. This emphasizes the importance of the placement of these discrete sensors, which is why a participant specific sensing insole was manufactured, placing sensors at the exact anatomical location of the boney landmarks, where peak loading is expected.

Although this paper presents the shear stress sensor sensitivities to calibration loading area and calibration loading locations for this sensor it is likely that these observations are true for other embedded in-shoe shear stress sensors. Other researchers measured in-shoe peak shear stresses from gait varied from 9 kPa to 140 kPa and calibration loading area varied from 20 mm diameter area (314 mm$^2$) –10,000 mm$^2$ (up to half the insole, approximated from the experimental Fig 3 in the paper as there was insufficient detail to give conclusive information on the loading area used) [5, 15]. It is likely that these variations in measurements are not due to inherent sensor inaccuracy or participant gait differences but likely to stem from calibration method differences. To the authors' knowledge, calibration loading area has not been investigated in other published studies, but it is suggested that calibration should be considered for all future in-shoe shear stress measurements.

## Human plantar loading consideration for sensor calibration

Fig 7 shows that calibration loading indenter diameters should be 20 mm and 40 mm for the hallux and both the first metatarsal head and the calcaneus respectively. However, due to limitations on sensor sensitivity beyond 30 mm from the center of the sensor a 30 mm indenter diameter was chosen for the first metatarsal head and calcaneus. These choices of calibration indenter diameters were determined from the comparison of the bench top testing normal stress profiles of different indenter diameters, with the participants' measured normal stress profile during walking. The bench top test showed that all the indenters resulted in normally distributed normal stress profile curves (Fig 7A), increasing in curve width with increasing

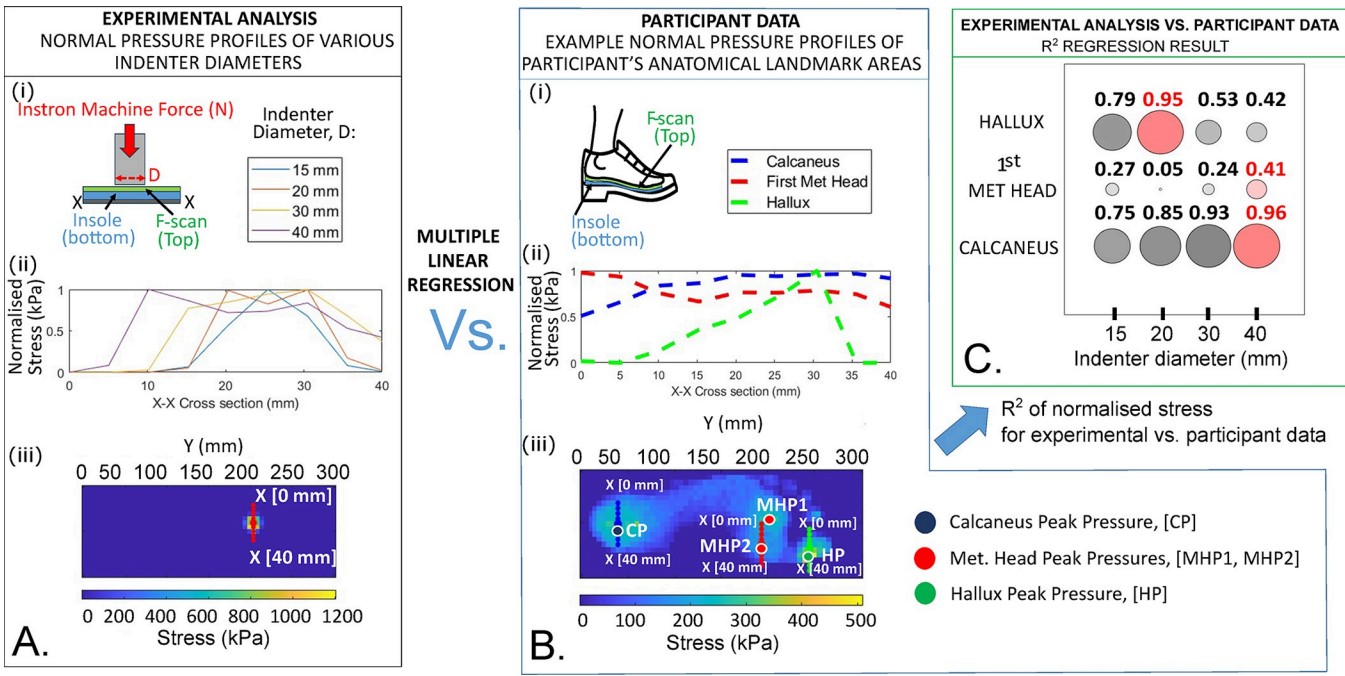

**Fig 7. Multiple linear regression analysis between the mean normal pressure profiles of participants' walking data, with normal pressure profiles of an experimental benchtop mechanical test of indenters of different diameters.** [A] Experimental normal pressure profiles: (i) Indenter experimental setup, (ii) Normal pressure profile curves width increases with increasing indenter diameter, (iii) F-scan pressure result that shows the cross section used to obtain these values used in ii. [B] Participant pressure profiles: (i) In-shoe gait lab experimental setup (ii) An example of participant's pressure profile over 20 gait cycles, showing the three normal pressure profiles of the foot at the calcaneus, first met head and hallux, (iii) F-scan pressure result that shows the cross section used to obtain the values. The image also shows the peaks for these three regions (Calcaneus peak CP, Hallux Peak HP and the metatarsal peaks MHP1 and MHP2). [C] Graphical representation of the regression analysis' coefficient of determination (or R-squared) results. Larger circles indicate a higher R-squared value, and red circles indicate the maximum R-squared in the sensor group. R-squared values are shown above the circles, and maximum is indicated as red font.

indenter diameters, reflecting a larger contact area of the applied force. An increasing curve width is also expected for the normal stress profiles of anatomical bones with increasing diameters (first metatarsal head ~15 mm, hallux ~20 mm, and calcaneus ~ 40 mm [31–36, 38]). The participants' measured normal stress for the hallux and the calcaneus regions of the foot had normal pressure distribution profiles that reflected their anatomical sizes, however, the presence of the second close metatarsal bone influenced the normal stress profile in the first metatarsal head area and widened the normal stress profile, more than what is expected from its anatomical diameter of ~15 mm (Fig 7B). The $R^2$ results of the multiple linear regression reflected this (Fig 7C), as the first metatarsal head correlates to the indenter size of 40 mm diameter. The $R^2$ value of the metatarsal head, however, is small at 0.41, indicating that there may be variability in the pressure distributions in that area, likely from gait variability within a participant's walk or between participants. The hallux and calcaneus regions of the foot have a normal pressure distribution profile that reflects the loading of the anatomical bones clearly ($R^2 \geq 0.95$) and can be matched with an indenter of a similar size to give a representative loading for calibration of 20 mm and 40 mm respectively. However, loading area results from Fig 6A show that sensor sensitivity converges for indenter areas greater than 25–30 mm diameter. Therefore, calibration indenter diameters were reduced to 30 mm for the first metatarsal head and calcaneus.

The implications of this for the SSS sensor are that calibration indenter sizes should be between 10–30 mm dependent on expected shear stress application areas. This finding is likely to be true for other embedded in-shoe shear stress sensors in the literature. The limitation

from this finding is that to obtain accurate shear stress measurements the user must know something about the shear stress loading profile which may be unknown. A possible way to mitigate for this may be to calibrate the sensor for a range of loading areas and to use a normal stress sensor to determine which indenter calibration area to use in post-processing.

## Shear sensor calibration and bench top mechanical test validation

The SSS sensor was highly accurate and repeatable when compared against the bench top mechanical test as seen in Fig 8. Results from Table 3 show that calibration error was insignificant with the mean absolute error (MAE) over the entire cycle in calibration < 0.00007 kPa for all magnitudes of loading, and errors at peak loading were < 5.8%.

The errors in the validation of the sensors at loading conditions different from the calibration were higher, but still showed a high accuracy for the sensors. The sensor was most accurate for low–medium shear stress magnitudes with up to <1.8 kPa for MAE, and < 8.7% for

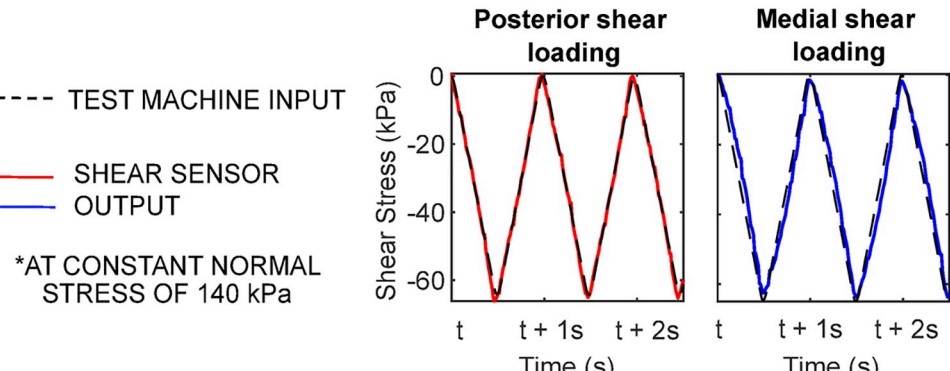

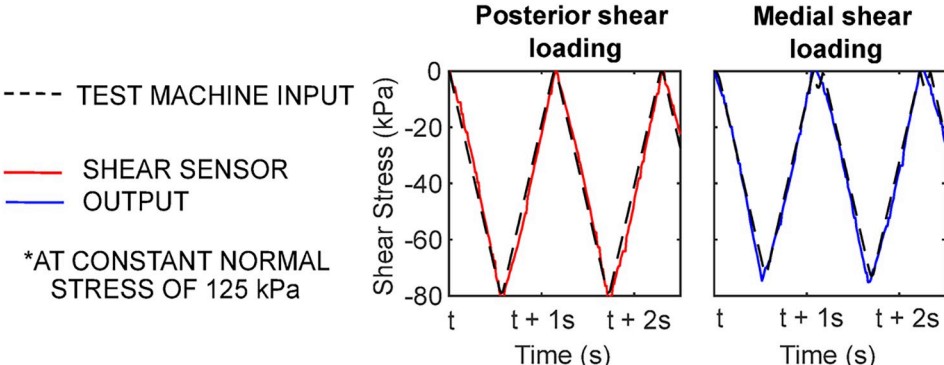

**Fig 8. Shear sensor benchtop mechanical testing calibration and validation results, sampled for 2 seconds of the cyclic loading.** [A] Sensor calibration for both anterior-posterior (AP) and medial-lateral (ML) directions at a 'medium' level of posterior and medial shear loading of 1 Hz cyclic loading of up to 70 kPa shear stress, at a constant normal stress of 140 kPa. [B] Sensor validation test result at medium level of shear cyclic loading (up to 70 kPa), at a different loading frequency (~0.85 Hz) and different constant normal stress (125kPa). All results for the different configurations of loading are shown in Table 3.

**Table 3. Sensor error, repeatability and range for calibrated measurements and validation results in different loading conditions.**

| LOADING CONFIGURATIONS | | | | CALIBRATION | | | LOADING CONFIG. | | VALIDATION | | |
|---|---|---|---|---|---|---|---|---|---|---|---|
| Load Direction | Measurement Axes and Frequency (Hz) | Applied Shear Stress* | Normal Stress (kPa) | Mean Absolute Error (kPa) | Peak Error** (%) | Repeatability (%) | Measurement Axes and Frequency (Hz) | Normal Stress (kPa) | Mean Absolute Error (kPa) | Peak Error** (%) | Repeatability (%) |
| **Medial** | ML, 1 Hz | LOW | 28 | 0.31 | 5.8 | 97.8 | ML, 0.85 Hz | 56 | 1.2 | 6.7 | 98.6 |
| | | MED. | 145 | $3.2 \times 10^{-5}$ | 1.5 | 99 | | 125 | 1.8 | 8.7 | 98.7 |
| | | HIGH | 280 | $7.1 \times 10^{-5}$ | 2.7 | 98.6 | | 400 | 17.3 | 22.4 | 96.6 |
| **Posterior** | AP, 1 Hz | LOW | 28 | $3.3 \times 10^{-5}$ | 2.3 | 98.4 | AP, 0.85 Hz | 56 | 0.4 | 1.1 | 98.6 |
| | | MED. | 145 | $2.4 \times 10^{-5}$ | 3.9 | 99 | | 125 | 0.9 | 4.4 | 99 |
| | | HIGH | 280 | $6.5 \times 10^{-5}$ | 1.8 | 99.1 | | 400 | 13.4 | 21.4 | 97.8 |
| **Clockwise 45˚ from Anterior** | | LOW | ***9.9 | | | | ML, 0.5 Hz | 60 | 0.8 | 11.5 | 97 |
| | | | ***9.9 | | | | AP, 0.5 Hz | | 1.4 | 10.9 | 98.5 |

*LOW = 0–14 kPa; MED. = 0–70 kPa; HIGH = 0–140 kPa

** Error at peak cyclic loading

*** Resultant stress in the measurement axes. Stresses rounded to nearest kPa.

error at peak loading (see example of medium magnitude measurements in Fig 8). Followed by the measurements at a resultant loading angle of 45˚ clockwise from the anterior direction (MAE <1.4 kPa; <11.5% peak error). These small errors could be attributed to errors in the validation setup, as an error of ± 5˚ would correspond to a peak shear stress error of up to 4.6%. The SSS sensor also showed good repeatability for all loading conditions (>97% repeatability in calibration and >96% repeatability in validation).

The highest errors in validation were at high shear stress magnitudes, over the expected plantar shear stress from gait, these were MAE <17.3 kPa and peak error <22.4%. This was likely due to the mechanical coupling of the high normal stress, pushing the total material deformation higher up the hyperelastic stress-strain curve of the sensor material (Fig 2D). At this region of the stress-strain curve, very small strains relate to high changes in stress making the SSS sensor more prone to measurement errors. However, the maximum errors translate to an error of ± 31.3 kPa, which is within the standard deviation of most plantar stress measurements from the literature of ± 50 kPa for shear stress [1–5, 15].

## Treadmill walking validation

For treadmill walking the SSS sensors measured the magnitude of shear stresses between 66.5 kPa—152.6 kPa in the AP direction, and 28.4 kPa– 128 kPa in the ML direction, full results are shown in Table 4. As expected, the ML shear range was lower than the AP shear range, as loading was expected to be predominantly in the AP direction. Loads were cyclic going from zero to peak value with the same frequency as gait which were at speeds of 0.92 and 0.95 ms$^{-1}$ for the healthy participant and participant with diabetes respectively. The only notable differences were in the direction of some of the peak plantar shear stresses.

No significant differences between both participants peak plantar stress values were observed (t-test of mean peak plantar stresses PPS, p>0.36, p>0.58 and p>0.57). This was expected, as both participants had a similar walking speed (0.92–0.95 ms$^{-1}$, and weight per insole area 32.4–35 kPa). However, this study aimed to demonstrate the feasibility, accuracy,

Table 4. Peak plantar stresses for both participants and and repeatability investigations for both intra and inter walks.

| | FOOT | Sensor Location | Peak Plantar Stress (kPa, mean + std. dev) | | | Intra-walk PPS percentage difference (%) | | | Inter-walk PPS percentage difference (%) | | |
|---|---|---|---|---|---|---|---|---|---|---|---|
| | | | Normal* | Shear AP** | Shear ML** | Normal | Shear AP | Shear ML | Normal | Shear AP | Shear ML |
| **Participant with Diabetes** | Right | 1st Met. Head | 199 ± 35 | 100 ± 22 | 73 ± 15 | 4 | 4 | 2 | 3 | 15 | 16 |
| | | Hallux | 238 ± 50 | -121 ± 20 | -61 ± 25 | 8 | 2 | 1 | 16 | 5 | 25 |
| | | Calcaneus | 323 ± 23 | 97 ± 23 | -128 ± 32 | 5 | 6 | 2 | 2 | 10 | 33 |
| | Left | 1st Met. Head | 163 ± 52 | 75 ± 9 | 28 ± 8 | 1 | 6 | 12 | 29 | 3 | 1 |
| | | Hallux | 213 ± 39 | 73 ± 7 | 111 ± 10 | 1 | 5 | 21 | 47 | 22 | 11 |
| | | Calcaneus | 316 ± 12 | 153 ± 7 | 102 ± 24 | 3 | 7 | 14 | 8 | 15 | 11 |
| **Healthy Participant** | Right | 1st Met. Head | 203 ± 25 | 66 ± 4 | 97 ± 13 | 2 | 4 | 9 | 8 | 17 | 2 |
| | | Hallux | 223 ± 20 | 116 ± 14 | 70 ± 4 | 5 | 5 | 2 | 6 | 37 | 10 |
| | | Calcaneus | 349 ± 11 | 137 ± 38 | 48 ± 15 | 2 | 14 | 11 | 7 | 9 | 22 |
| | Left | 1st Met. Head | 141 ± 32 | 78 ± 16 | 75 ± 6 | 5 | 13 | 9 | 13 | 4 | 6 |
| | | Hallux | 260 ± 64 | 75 ± 19 | 54 ± 17 | 8 | 13 | 10 | 22 | 16 | 17 |
| | | Calcaneus | 341 ± 10 | 101 ± 15 | 86 ± 10 | 8 | 6 | 5 | 4 | 12 | 2 |
| | | **Intra-/Inter- walk (peak % difference) Mean ± Std. dev** | | | | 4 ± 3 | 7 ± 4 | 8 ± 6 | 6 ± 5 | 7 ± 3 | 7 ± 2 |

*Normal plantar stress measured from FlexiForce sensor

** Shear plantar values are signed to a direction, i.e. Posterior = -ve; Medial = -ve

Stresses are rounded up to nearest kPa. Intra-walk: investigation within 15 minute walk; Inter-walk: investigation between two 15 minute walks.

and repeatability of the SSS system so no conclusions should be drawn on plantar stress for general people with diabetes and healthy populations for this study.

The shear measurements of the SSS sensor was highly repeatable when comparing data recorded for both within the 15-minute treadmill walk (intra-walk), and between the two 15-minute walks (inter-walk). The mean and standard deviation of the percentage difference of peak plantar stresses were $\leq 8\% \pm 6\%$ for both investigations. Intra-walk differences were lower than inter-walk–with the highest percentage difference of 21% measured by the SSS sensor for the ML Shear (Hallux, Left foot, participant with diabetes). Other measurements from the shear stress sensors were < 15% difference. For inter-walk, the highest PPS percentage difference was measured by the commercial Flexiforce sensor of 47% difference in normal stress (Hallux, left foot, participant with diabetes), followed by 37% for the AP shear of the SSS sensor (Hallux, right, healthy) and 33% for the ML shear of the SSS sensor (Calcaneus, right, participant with diabetes).

## Conclusion

### Calibration and material coupling for shear stress sensors

To the author's knowledge, this study is the first to address in-shoe shear sensing material coupling and unexplored complexities in calibration for shear sensing. The results illustrate that due to sensor and material coupling with adjacent structures the area which contributes to the measured shear can be larger than the area of the sensor. This has important implications for shear sensor calibration, firstly in terms of the location of the sensor and the anatomical region that is to be measured, and secondly in terms of the indenter area used for calibration. These results have significance for all researchers developing systems to measure in-shoe plantar shear stress as these factors will affect the magnitude of shear sensed. Furthermore, these results may partially explain the variation in magnitudes of shear measured at the same anatomical locations by different researchers. A suggested approach for shear sensor calibration is

shown below (for detail see methods 'Human plantar loading specific sensor calibration' section):

- *Determine the sensing area*: Material coupling between the shear sensor and adjacent regions can result in the area sensed being greater than then actual area of the sensor.

- *Determine the distribution of plantar loading*: Normal stress distribution will be indicative of shear stress distribution, whilst foot anatomy, for example the hallux, will determine the loading area.

- *Decision for calibration indenter area*: Informed by both the sensing area and the distribution and magnitude of plantar loading.

## Developed shear stress system sensor

**Sensor performance.** A novel Shear Stress System (SSS) sensor composed of a strain gauge rosette, normal pressure sensor and stiffener to concentrate loading at the desired sensor location and mitigate against material coupling was developed and evaluated. Sensor locations were anatomically matched and measured the plantar loading profiles to inform calibration of each sensor at a specific location. This study conducted a thorough experimental validation of the shear sensor through mechanical bench top testing and with human participant treadmill walking. Shear sensing results demonstrated high repeatability (>97%) and high accuracy in the expected measurement range for plantar shear stress (mean absolute errors < ±2 kPa) with error increasing for very high shear stresses (mean absolute errors < ±17 kPa) compared to bench top mechanical tests and repeatability for treadmill walking of 15-minutes duration with less than 21% variability within walking, and less than 37% variability between walks (which was lower than the commercial normal pressure sensors of 47% used in this study).

**Limitations.** A rosette strain gauge was chosen for determining unknown principal directions, however it restricted complete strain separation in the AP and ML directions. For exclusive separation, a 0˚–90˚ strain gauge in the ML and AP axes could be adopted. The manual assembly of the sensors and alignment of the sensor in relation to the AP and ML directions affect shear measurement. This has been controlled through careful manufacture, but some small errors will remain. The chosen alignment of the strain gauge rosette in the ML direction was to reduce the fatigue on the soldered joints, this resulted in a decreased sensitivity in the AP direction due to the 45˚ off-alignment of the gauges with this axis.

Relative stiffness of the silicone and the strain gauge rosette will affect strain transfer between the two materials. Material properties of the silicone is highly important for measurement accuracy, sensitivity, and range, and warrants further investigation.

**Future work.** A three-part linear fitting procedure was adopted to calibrate the SSS sensor accommodating the hyperelastic material properties, in the future consideration of alternative fits to capture viscoelastic effects could be made. Despite observing minimal shear sensor temperature response, variability between 20–30˚C, literature indicates foot temperatures may be as high as 35˚ in people with diabetes [39, 40], this should be considered in the future. In this proof-of-concept study, the size of calibration area was based on average pressure profiles, a suitable assumption with little participant variation. However, future larger studies may require participant-specific calibration to address varying loading profiles, particularly due to gait variability.

## Author Contributions

**Conceptualization:** Athia H. Haron, Helen Dawes, Glen Cooper, Andrew Weightman.

**Data curation:** Athia H. Haron, Helen Dawes, Andrew Weightman.

**Formal analysis:** Athia H. Haron, Glen Cooper, Andrew Weightman.

**Funding acquisition:** Helen Dawes, Glen Cooper, Andrew Weightman.

**Investigation:** Athia H. Haron, Lutong Li, Jiawei Shuang, Chaofan Lin, Helen Dawes, Maedeh Mansoubi, Damian Crosby, Garry Massey, Glen Cooper, Andrew Weightman.

**Methodology:** Athia H. Haron, Lutong Li, Jiawei Shuang, Chaofan Lin, Maedeh Mansoubi, Damian Crosby, Glen Cooper, Andrew Weightman.

**Project administration:** Athia H. Haron, Glen Cooper, Andrew Weightman.

**Resources:** Helen Dawes, Neil Reeves, Frank Bowling, Andrew Weightman.

**Software:** Athia H. Haron, Damian Crosby, Garry Massey.

**Supervision:** Helen Dawes, Neil Reeves, Frank Bowling, Glen Cooper, Andrew Weightman.

**Validation:** Athia H. Haron, Helen Dawes, Glen Cooper, Andrew Weightman.

**Visualization:** Athia H. Haron, Glen Cooper, Andrew Weightman.

**Writing – original draft:** Athia H. Haron, Glen Cooper, Andrew Weightman.

**Writing – review & editing:** Athia H. Haron, Lutong Li, Jiawei Shuang, Chaofan Lin, Helen Dawes, Maedeh Mansoubi, Damian Crosby, Garry Massey, Neil Reeves, Frank Bowling, Glen Cooper, Andrew Weightman.

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
