## [Decision Letter · Decision Letter 0]

12 Mar 2024

PONE-D-24-01745In- Shoe Plantar Shear Stress Sensor Design, Calibration and Evaluation for the Diabetic Foot.PLOS ONE

Dear Dr. Weightman,

Thank you for submitting your manuscript to PLOS ONE. After careful consideration, we feel that it has merit but does not fully meet PLOS ONE’s publication criteria as it currently stands. Therefore, we invite you to submit a revised version of the manuscript that addresses the points raised during the review process.

**ACADEMIC EDITOR: **The authors provided a study presenting the design of a shear stress sensor for diabetic foot monitoring. This study was evaluated by two experts in the field who acknowledged the significance of the research. However, they also highlighted several major and minor issues that need addressing before the manuscript can be accepted. Additionally, based on my expertise in the field, I observed a deficiency in the potential application of such a sensor concerning the data it provides. For this reason, I suggest that the authors also review recent literature that could bolster the sensor's applicability in diagnostic schemes. The pressure information related to diabetic feet can be particularly valuable in posturography, as suggested by Mengarelli, Alessandro, et al. in "Multiscale Fuzzy Entropy Analysis of Balance: Evidences of Scale-Dependent Dynamics on Diabetic Patients with and without Neuropathy," published in IEEE Transactions on Neural Systems and Rehabilitation Engineering 31 (2023): 1462-1471.

We look forward to receiving your revised manuscript.

Kind regards,

Andrea Tigrini, Ph.D.

Academic Editor

PLOS ONE

“This work was partially funded by Engineering and Physical Sciences Research Council (EPSRC) grant number EP/W00366X/1”

Additional Editor Comments:

The authors provided a study presenting the design of a shear stress sensor for diabetic foot monitoring. This study was evaluated by two experts in the field who acknowledged the significance of the research. However, they also highlighted several major and minor issues that need addressing before the manuscript can be accepted. Additionally, based on my expertise in the field, I observed a deficiency in the potential application of such a sensor concerning the data it provides. For this reason, I suggest that the authors also review recent literature that could bolster the sensor's applicability in diagnostic schemes. The pressure information related to diabetic feet can be particularly valuable in posturography, as suggested by Mengarelli, Alessandro, et al. in "Multiscale Fuzzy Entropy Analysis of Balance: Evidences of Scale-Dependent Dynamics on Diabetic Patients with and without Neuropathy," published in IEEE Transactions on Neural Systems and Rehabilitation Engineering 31 (2023): 1462-1471.

Reviewers' comments:

Reviewer's Responses to Questions

**Comments to the Author**

1. Is the manuscript technically sound, and do the data support the conclusions?

Reviewer #1: Yes

Reviewer #2: Yes

2. Has the statistical analysis been performed appropriately and rigorously? 

Reviewer #1: Yes

Reviewer #2: Yes

3. Have the authors made all data underlying the findings in their manuscript fully available?

Reviewer #1: Yes

Reviewer #2: Yes

4. Is the manuscript presented in an intelligible fashion and written in standard English?

Reviewer #1: Yes

Reviewer #2: Yes

5. Review Comments to the Author

Reviewer #1: The Authors presented a work aimed at proposing the design and calibration of an in-shoe plantar shear stress sensor for applications in the assessment of diabetic foot ulceration. Specifically, focus is made on the evaluation in terms of sensor loading area and location. The manuscript is generally well written, but needs improvement. My major concern is about the description of the performed experiments in the methodology section. Indeed, the subdivision followed to describe calibration procedure and validation procedure is confusing thus not allowing the optimal fruition on what steps have been followed (e.g., the gait laboratory treadmill walking test is introduced at line 236 but participants are mentioned also in previous sections in the text). Providing an outline of the methodology content and/or improving subsections’ subtitles may be of help. Further minor comments are provided in the following.

- How were the compressive strains values for ε1 and ε2 determined? Please provide a brief sentence.

- In Fig. 2 label, “SSS” acronym is not defined. Same for “DAQ” in Fig. 3. Figures' labels should be self explaining.

- Line 185: call to the figure is not clear.

- Line 205-207: the sentence is not clear, please rephrase. Some interrupted sentences are present in the text.

- The reported state of art literature regarding the context in which the presented study may find application is very specific. I suggest the Authors to broaden the research context description by mentioning more in general where is the state of art for DFU risk prediction, in order to help the more general part of the Journal’s readership to become aware on the strength of the proposed work. Regarding to this, a recent review on “artificial intelligence methodologies applied to technologies for screening, diagnosis and care of the diabetic foot” may provide insightful information on which quantitative measures are suggested for an early identification of DFU risk, including plantar-based measurements. Indeed, in my opinion the quantitative information provided by shear stress measurement may also be used in artificial intelligence based approaches for the prediction of diabetic foot ulceration.

Reviewer #2: Manuscript title: In- Shoe Plantar Shear Stress Sensor Design, Calibration and Evaluation for the Diabetic Foot.

This work investigated the pressing issue of Diabetic Foot Ulceration (DFU), a significant global healthcare challenge. It highlights the crucial need for technology to identify the onset of DFU, enabling timely intervention, improving healthcare delivery, and reducing economic costs associated with its management. Specifically, the study focuses on the role of high-foot plantar stresses, including shear stress, in the formation of DFUs. It acknowledges the challenges in accurately measuring shear stress and the limited research in this area. It's commendable that the paper is well-written.

comments

• The abstract should emphasize the novelty of the research findings.

• The introduction section is crucial, but the author hasn't given it sufficient attention. It should include a discussion of currently available research.

• Abbreviations should be explained at the beginning. In the Methods section, there is confusion regarding the symbols used.

• The author should explain the effect of silicone stiffness on the strain gauge rosette.

• More detailed explanations of the experimental setup and testing methods are needed, including the number of subjects and their body parameters.

6. PLOS authors have the option to publish the peer review history of their article (what does this mean?). If published, this will include your full peer review and any attached files.

Reviewer #1: No

Reviewer #2: **Yes: **Akhil V M

---

## [Author Response · Author response to Decision Letter 0]

25 Jun 2024

We thank the reviewers for considering our manuscript and for providing their supportive feedback. We are pleased that the reviewers agree our work addresses the importance of creating a technology capable of measuring the full stress state of the diabetic foot to prevent ulceration and the accompanying challenges in measuring shear stress in a reliable and accurate manner. 

Please see our response to reviewer document attached for the detailed replies to comments.

Kind regards,

Prof. Andrew Weightman

---

## [Decision Letter · Decision Letter 1]

16 Jul 2024

PONE-D-24-01745R1In- Shoe Plantar Shear Stress Sensor Design, Calibration and Evaluation for the Diabetic Foot.PLOS ONE

Dear Dr. Weightman,

Thank you for submitting your manuscript to PLOS ONE. After careful consideration, we feel that it has merit but does not fully meet PLOS ONE’s publication criteria as it currently stands. Therefore, we invite you to submit a revised version of the manuscript that addresses the points raised during the review process.

**ACADEMIC EDITOR: **Authros addressed the majority of the points.  However, some minor concerns still need revision.

We look forward to receiving your revised manuscript.

Kind regards,

Andrea Tigrini, Ph.D.

Academic Editor

PLOS ONE

Journal Requirements:

**Additional Editor Comments:**

Authros addressed the majority of the points. However, some minor concerns still need revision.

Reviewers' comments:

Reviewer's Responses to Questions

**Comments to the Author**

1. If the authors have adequately addressed your comments raised in a previous round of review and you feel that this manuscript is now acceptable for publication, you may indicate that here to bypass the “Comments to the Author” section, enter your conflict of interest statement in the “Confidential to Editor” section, and submit your "Accept" recommendation.

Reviewer #1: (No Response)

Reviewer #2: All comments have been addressed

2. Is the manuscript technically sound, and do the data support the conclusions?

Reviewer #1: Yes

Reviewer #2: Yes

3. Has the statistical analysis been performed appropriately and rigorously? 

Reviewer #1: Yes

Reviewer #2: N/A

4. Have the authors made all data underlying the findings in their manuscript fully available?

Reviewer #1: Yes

Reviewer #2: Yes

5. Is the manuscript presented in an intelligible fashion and written in standard English?

Reviewer #1: Yes

Reviewer #2: Yes

6. Review Comments to the Author

Reviewer #1: The Authors responded to almost all Reviewers' comments, but information on anthropometric characteristics and body parameters is not yet reported, thus compromising reproducibility. Furthermore, the authors should prefer to use “people with diabetes” or similar, instead of “diabetics”. Even if the effort made to improve the explanation of the methodology is recognisable, Figure 1 should still be referred to in the text wherever necessary (perhaps adding letters to the panels could help this purpose). The resolution of Figure 1 should also be improved.

Reviewer #2: The authors have thoroughly addressed all the comments, demonstrating a positive and constructive approach. They have provided detailed responses and made the necessary revisions to improve the manuscript, ensuring that all concerns and suggestions from the reviewers have been adequately considered and integrated.

7. PLOS authors have the option to publish the peer review history of their article (what does this mean?). If published, this will include your full peer review and any attached files.

Reviewer #1: No

Reviewer #2: **Yes: **AKHIL V M

---

## [Author Response · Author response to Decision Letter 1]

8 Aug 2024

Response to Reviewers 17-07-2024

Authors’ General Response:

We thank the all reviewers for their thorough and constructive review of our paper, and are pleased to attach the following minor amendments to the paper as suggested

Reviewer 1 Comment #1: The Authors responded to almost all Reviewers' comments, but information on anthropometric characteristics and body parameters is not yet reported, thus compromising reproducibility. 

Author’s Response: Thank you for your comment, we appreciate that this is important information for reproducibility.

We have stated the following anthropomorphic parameters at the start of the ‘sensor validation section’ as follows: ‘To further validate the sensors, a gait laboratory treadmill walking test was performed on a single anthropometrically matched healthy participant and a single diabetic participant (both male and 45 years old, weighing 88 kg and 75 kg, EU shoe size 44 and 42, weight per insole area 32 kPa and 35 kPa, walking speed 0.92 ms-1 and 0.95 ms-1 for the healthy and diabetic respectively)…’. These parameters were disclosed as they were similar parameters described in the other similar study in PLOS ONE of measuring in-shoe shear (Tang et. al, 2023). We have added height details to match this paper as follows:

Manuscript Edit (page 15, line 262-265): ‘To further validate the sensors, a gait laboratory treadmill walking test was performed on a single anthropometrically matched healthy participant and a single diabetic participant (both male and 45 years old, weighing 88 kg and 75 kg, height of 1.75 and 1.66 m, EU shoe size 44 and 42, weight per insole area 32 kPa and 35 kPa, walking speed 0.92 ms-1 and 0.95 ms-1 for the healthy participant and participant with diabetes respectively)

Reviewer 1 Comment #2: Furthermore, the authors should prefer to use “people with diabetes” or similar, instead of “diabetics”. 

Author’s Response: Thank you for your comment, we agree that this is preferable. As such, we’ve changed this on the following sections/sentences in the manuscript (italicized):

Abstract Edit: ‘…The sensing insole, coupled with the calibration procedure, was tested one participant with diabetes and one healthy participant during two sessions of 15 minutes of treadmill walking. Calibration with different indenter areas (from 78.5 mm2 to 707 mm2) and different positions…’

Manuscript Edit (page 3, line 41): ‘Diabetic foot ulceration (DFU) affects 15-25% of people with diabetes at some point in their lifetime [1]’

Manuscript Edit (page 3, line 52-53): ‘Contradictory results are typical from these studies using custom-built shear stress measurement devices due to the relatively low numbers of participants with diabetes tested in the trials, …’

Manuscript Edit (page 3, line 55-56): ‘Larger scale studies with matched control groups are required to provide firm conclusions on plantar surface shear stresses experienced by people with diabetes.’

Manuscript Edit (page 12, line 264-267): ‘To further validate the sensors, a gait laboratory treadmill walking test was performed on a single anthropometrically matched healthy participant and a single diabetic participant (both male and 45 years old, weighing 88 kg and 75 kg, height of 1.75 and 1.66 m, EU shoe size 44 and 42, weight per insole area 32 kPa and 35 kPa, walking speed 0.92 ms-1 and 0.95 ms-1 for the healthy participant and participant with diabetes respectively).

Manuscript Edit (page 13, line 279-280): ‘…designed with shock-absorbing properties and a larger volume, ideal for people with diabetes.’

Manuscript Edit (page 19, line 402-403): ‘Loads were cyclic going from zero to peak value with the same frequency as gait which were at speeds of 0.92 and 0.95ms-1 for the healthy participant and participant with diabetes respectively.’

Manuscript Edit (page 20, column 1, Table 4)

Manuscript Edit (page 21, line 410-412): ‘However, this study aimed to demonstrate the feasibility, accuracy, and repeatability of the SSS system so no conclusions should be drawn on plantar stress for general people with diabetes and healthy populations for this study.’

Manuscript Edit (page 21, line 415-420): ‘Intra-walk differences were lower than inter-walk – with the highest percentage difference of 21% measured by the SSS sensor for the ML Shear (Hallux, Left foot, participant with diabetes). Other measurements from the shear stress sensors were < 15% difference. For inter-walk, the highest PPS percentage difference was measured by the commercial Flexiforce sensor of 47% difference in normal stress (Hallux, left foot, participant with diabetes), followed by 37% for the AP shear of the SSS sensor (Hallux, right, healthy) and 33% for the ML shear of the SSS sensor (Calcaneus, right, participant with diabetes).’

Manuscript Edit (page 23 line 462-464): ‘..Despite observing minimal shear sensor temperature response, variability between 20–30 °C, literature indicates foot temperatures may be as high as 35° in people with diabetes [39, 40]..’

Reviewer 1 Comment #3: Even if the effort made to improve the explanation of the methodology is recognisable, Figure 1 should still be referred to in the text wherever necessary (perhaps adding letters to the panels could help this purpose). The resolution of Figure 1 should also be improved.

Author’s Response: Thank you for your comment, we have included, where necessary, references to Figure 1 in the main manuscript as suggested and added letters to panels as well as improved the quality of the figure. 

Manuscript Edit (page 8 line 172-173): To investigate the effect of calibration on the sensor’s performance, both shear and normal force were applied to the SSS sensor insole (summarised in Fig 1A).

Manuscript Edit (page 11 line 231-232): An illustrated summary of this investigation can also be found in Fig 1B.

Manuscript Edit (page 14 line 244): The following section describes the sensor validation, as summarised in Fig 1C.

Reviewer 2 comment: The authors have thoroughly addressed all the comments, demonstrating a positive and constructive approach. They have provided detailed responses and made the necessary revisions to improve the manuscript, ensuring that all concerns and suggestions from the reviewers have been adequately considered and integrated.

Authors’ Response: Thank you for reviewing our study and providing helpful feedback to improve our paper, we greatly appreciate this. We note from your comment that you are satisfied with our edits in response to your feedback.

Other amendments:

After a resubmission review of the manuscript, we have also identified a necessary amendment and have made this edit in the manuscript as follows (italicised):

Manuscript Edit (page 15, line 326-329): ‘Other researchers measured in-shoe peak shear stresses from gait varied from 9 kPa to 140 kPa and calibration loading area varied from 20 mm diameter area (314 mm2) –10,000 mm2 (up to half the insole, approximated from the experimental figure 3 in the paper as there was insufficient detail to give conclusive information on the loading area used) [5,15].’

---

## [Editor Report · Decision Letter 2]

15 Aug 2024

In- Shoe Plantar Shear Stress Sensor Design, Calibration and Evaluation for the Diabetic Foot.

PONE-D-24-01745R2

Dear Dr. Weightman,

We’re pleased to inform you that your manuscript has been judged scientifically suitable for publication and will be formally accepted for publication once it meets all outstanding technical requirements.

Kind regards,

Andrea Tigrini, Ph.D.

Academic Editor

PLOS ONE

Additional Editor Comments (optional):

The authors have addressed all comments, and the manuscript is now ready for publication.
---

## [Editor Report · Acceptance letter]

23 Aug 2024

PONE-D-24-01745R2 

PLOS ONE

Dear Dr. Weightman, 

I'm pleased to inform you that your manuscript has been deemed suitable for publication in PLOS ONE. Congratulations! Your manuscript is now being handed over to our production team.

Kind regards, 

on behalf of

Dr. Andrea Tigrini 

Academic Editor

PLOS ONE